

# Deep Learning Model based on Multi-scale Feature Fusion for Precipitation Nowcasting

Jinkai Tan[1,2,*], Qiqiao Huang[1,*], and Sheng Chen[1,3,□]

[1]Northwest Institute of Eco-Environment and Resources, Chinese Academy of Sciences, Lanzhou 730000, China
[2]School of Atmospheric Sciences, and Key Laboratory of Tropical Atmosphere-Ocean System (Ministry of Education), Sun Yat-sen University, Zhuhai 519082, China
[3]Southern Marine Science and Engineering Guangdong Laboratory (Zhuhai), Zhuhai 519082, China
[*]These authors contributed equally to this work.

**Correspondence:** Sheng Chen (chensheng@nieer.ac.cn)

**Abstract.** Accurate forecast of heavy precipitation remains a challenging task in most deep learning (DL)-based models. This study proposes a novel DL architecture named 'Multi-scale Feature Fusion' (MFF) for precipitation nowcasting for a lead time of up to 3 h. The basic idea is to apply convolution kernels with various sizes to achieve multi-scale receptive fields and then capture the movement features of the precipitation system (e.g. shape, movement direction, and moving speed). Meanwhile,

the architecture implants the mechanism of discrete probability to reduce uncertainties and forecast errors, so that heavy precipitations can be produced even at longer lead time. The model uses four year's radar echo data from 2018 to 2021 for model training and one year's data of 2022 for model testing. The model is compared with three existing extrapolative models: time series residual convolution (TSRC), optical flow (OF), and UNet. Results show that MFF obtains relatively superior forecast skills with a high probability of detection (POD), low false alarm rate (FAR), small mean absolute error (MAE), and

high structural similarity index (SSIM). The most commendable result is that MFF can predict high-intensity precipitation fields at 3 h lead time while the other three models can not. Additionally, it can be found from the results of radially averaged power spectral (RAPS) that MFF shows improvement in the smoothing effect of the forecast field. Future works will pay more attention to multi-source meteorological variables, the structural adjustments of the network, and the combinations with numerical models to further improve the forecast skills of heavy precipitations at longer lead times.

## 1   Introduction

Heavy precipitation is a key driver of a variety of natural disasters, including floods, landslides, and mud-rock flows, which pose a threat to both life and property. The term 'nowcasting' refers to predicting precipitation over a certain region within a short time frame (typically less than 3 hours) and with a fine-grained spatiotemporal resolution (Ayzel et al., 2020; Czibula et

al., 2021). It is an attractive research topic in the field of hydrometeorology. The destruction of a precipitation event mainly





depends on its intensity, duration, and falling area. Therefore, accurate and timely nowcasting has become an indispensable link
for disaster early warning and emergency response (Chen et al., 2020; Ehsani et al., 2021). However, real-time, large-scale, and
fine-grained precipitation nowcasting remains a challenging task due to the inherent complexities of atmospheric conditions
(Ehsani et a., 2021; Kim et al., 2021).

Conventional approaches for precipitation nowcasting mainly include numerical weather prediction (NWP)-based methods
(Sun et al., 2014; Yano et al., 2018) and radar echo-based quantitative forecasts (Liguori et al., 2014). The NWP models describe
atmospheric phenomenons by solving a series of differential equations and thus predict precipitation dynamics (Dupuy et al.,
2021). They represent the main tools for precipitation forecasts. However, these models are computationally intensive, time-
consuming, and difficult to assimilate the local data, their forecast products depend on initial/boundary conditions (Marrocu

et al., 2020; Ehsani et al., 2021). Besides, the first few hours of precipitation predictions by NWP models are invalid so they
are not commonly used in nowcasting (Han et al., 2019; Yan et al., 2020). The radar echo-based quantitative models use
the so-called Z-R relationship (radar reflectivity 'Z' and precipitation intensity 'R') to drive precipitation rates and further
estimate precipitation accumulations. Especially, the optical flow (OF) model is the simplest technique in radar echo-based
quantitative forecast models. It consists of tracking and extrapolation, where an advection field is estimated from a series

of consecutive radar echo images. This field is then used to extrapolate recent radar echo images through semi-Lagrangian
schemes or interpolation procedures (Ayzel et al., 2019). Progress and achievements in precipitation nowcasting with variations
of the OF model have also been documented in many other studies (Marrocu et al., 2020; Pulkkinen et al., 2019; Ayzel et al.,
2019; Prudden et al., 2020; Liu et al., 2015; Woo et al., 2017; Li et al., 2018). Although the OF model and its variations achieved
great advances in precipitation nowcasting, they have certain limitations due to the assumption of a constant advection field

(Prudden et al., 2020; Li et al., 2021).

    Recently, deep learning (DL) techniques in precipitation nowcasting have drawn much attention from numerous studies,
due to their superior performances in tracking and processing successive frames of radar echo video/image. For example, Shi
et al. (2015) treated precipitation nowcasting as a spatiotemporal sequence predictive problem, and proposed convolutional
long short-term memory (ConvLSTM) architecture which helps to capture both spatial and temporal features of radar echo

sequences. This model outperformed the OF method. Considering the change of radar echo over time, in their follow-up study
(Shi et al., 2017), they introduced a trajectory GRU (TrajGRU) model which used the same convolutional and recursive net-
works as in the ConvLSTM, while the spatially-variant relationship of radar echo is excavated by its sub-networks. Moreover,
Chen et al. (2020) built a new architecture with a transition path (star-shaped bridge, SB) based on ConvLSTM which gleans
more latent features and makes the model more robust, the model was used in precipitation nowcasting over the Shanghai area

and achieved better performances than some conventional extrapolation methods. To improve the limitation of time-step re-
duction in the ConvLSTM model, Yasuno et al. (2021) proposed a rain-code approach with multi-frame fusion, thus the model
has a forecast lead time of 6 hours. Ronneberger et al. (2015) presented a deep network with U-shaped architecture, namely
U-Net, consisting of a contracting path to capture context and an expanding path that enables precise positioning. This model
was first used in biomedical segmentation applications. Numerous attempts to develop an UNet-based precipitation nowcast-

ing model and obtained certain success, such as the 'RainNet' in Germany (Ayzel et al., 2020), the 'MSDM' in eastern China



(Li et al., 2021), the 'Convolutional Nowcasting-Net' with IMERG products (Ehsani et al., 2021), the 'SmaAt-UNet' in the Netherlands (Trebing et al., 2021), the 'FURENet' for convective precipitation nowcasting (Pan et al., 2021), the nowcasting system with ground-based radars and geostationary satellites imagery (Lebedev et al., 2019), and Sadeghi et al., (2020) used UNet convolutional neural network and geographical information for improving near real-time precipitation estimation.

Apart from ConvLSTM-based and UNet-based models, many plug-and-play modules for radar-based nowcasting either trim deformable network architectures or implant various feature extraction operations into network architectures. For example, Ravuri et al. (2021) presented a conditional generative model for the probabilistic nowcasting which produced realistic and spatiotemporally consistent predictions with a lead time of up to 90 minutes. This model eliminate the blurry nowcasting maps and outperformed UNet and PySTEPS (Pulkkinen et al., 2019). The Google Research group (Sønderby et al., 2020) developed

a weather probabilistic model 'MetNet' which used axial self-attention mechanisms to unearth weather patterns from large-scale radar and satellite data, it provided probabilistic precipitation maps for up to 8 hours over the continental United States at a spatial resolution of 1 km2 and temporal resolution of 2 minutes. The Huawei Cloud group (Bi et al., 2022) devised a 3D earth specific transformer module and developed 'Pangu-Weather', a high-resolution system for the global weather forecast. The system showed good application prospects for its superior performance in many downstream forecast tasks such as wind,

temperature, and typhoon forecasts. Researchers from DeepMind and Google (Lam et al., 2022) proposed a novel machine learning weather simulator named 'GraphCast', which was an autoregressive model based on graph neural networks and a high-resolution multi-scale mesh representation. The model produced medium-range global weather forecasting for up to 10 days. The Microsoft Research group (Tung et al., 2023) developed and demonstrated the 'ClimaX' model which extended the Transformer architecture with novel encoding and aggregation blocks, the model resulted in superior performance on

benchmarks for both weather forecasting and climate projections. Similarly, the author of Chen et al. (2023) presented an advanced data-driven global medium-range weather forecast system named 'FengWu', which is equipped with model-specific encoder-decoders, cross-modal fusion Transformer, and a replay buffer mechanism, and it solved the medium-range forecast problem from a multi-modal and multi-task perspective. Marrocu et al. (2020) proposed the 'PreNet' model which is based on a widely-used semisupervised and unsupervised learning DL method named 'generative adversarial network' (GAN, Goodfellow

et al., 2014). The model's performance was compared with state-of-the-art OF procedures and shown remarkable superiority. Zheng et al. (2022) established the 'GAN–argcPredNet' model which was also based on GAN architecture, and it can reduce the prediction loss in a small-scale space and show more detailed features among prediction maps.

However, some limitations/challenges in the above DL-based models for precipitation nowcasting are widely reported. First, because precipitation dynamics are quite complex and DL models are difficult to extrapolate short-term local convection

or precipitation fields by learning the prior knowledge from historical radar echo data alone (Su et al., 2020; Chen et al., 2020; Ehsani et al., 2021), it is challenging to predict fast-moving precipitation systems or short-term local convections with rapid growth and dissipation. Second, accumulative errors and uncertainties usually occur during iterative forecasts due to the discrepancy between the model's training and testing process (Ayzel et al., 2020; Prudden et al., 2020; Li et al., 2020; Singh et al., 2021; Huang et al., 2023), resulting in low values of heavy precipitation, smoothing or blurry forecast fields. Third, the

convolution operation in DL models covers precipitation fields as comprehensively as possible but is unable to reveal the rapid





changes in echo intensity, deformation, and movement of precipitation fields (Ehsani et al., 2021; Kim et al., 2021). Therefore, DL models inevitably produce some undesirable forecast outputs, such as declining forecast skills with increasing lead time, smoothing and blurry precipitation fields, missing extreme precipitation events, and poor forecast skills for precipitation growth and dissipation.

Large-scale precipitation systems are affected by many factors such as prevailing westerlies, trade-wind zone, mesoscale weather systems, land-sea distributions, and topography effects (Huang et al., 2023; Luo et al., 2023). Therefore, accurate and real-time precipitation nowcasting is still a very challenging issue. In this study, we apply large-scale radar echo data and elaborately design a DL architecture named 'Multi-scale Feature Fusion' (MFF), which focuses on detecting radar echo multi-scale feature (e.g. intensity, movement direction, and speed), and is expected to improve forecast skills for precipitation

growth and dissipation, fast-moving precipitation systems and heavy precipitations. The rest of this article is organized as follows: Section 2 presents the data materials, the detailed method, and the framework of the model. Section 3 describes the experimental results including two precipitation cases, and discuss the advantages and disadvantages of the four models. Section 4 draws conclusions and explores some possible future works.

## 2    Materials, Methods, and Models

### 2.1    Radar Reflectivity Image Products

Weather radar is the main monitoring instrument for precipitation systems and severe convective weather such as hail, gale, tornado, and flash flood. As of November 2022, the China Meteorological Administration has deployed the China Next Generation Weather Radar (CINRAD) network composed of 236 C-band and S-band Doppler weather radars over China (see Fig. 1). The CINRAD network is distributed heterogeneously across China except in complex terrain (Min et al., 2019), and measures

the moving speeds of the meteorological target relative to radars and further inverts various types of meteorological products. This study collects and sorts out radar reflectivity image products of five seasons (March to August) from 2018 to 2022, its temporal resolution is 6 min and its coverage area is $(73^o E - 135^o E, 10^o N - 55^o N)$. The data pre-processing steps are as follows:

(i) Because radar echoes are affected by low-altitude objects (e.g. massif, building, tree, etc), so sham echoes are often

produced at low-elevation areas. Therefore, we firstly remove the anomalous radar echoes and detach the surplus annotations (e.g. city name, demarcation, and river) from each image. Secondly, to reduce the influence of sham echoes on the extrapolative model, we use a local-mean filter algorithm for radar image denoising, and then the radar reflectivities are transformed into precipitation values based on the Z-R relationship.

(ii) The extrapolative model will be difficult to converge due to the great numerical differences among each echo reflectivity.

Therefore, we normalize the initial radar reflectivities to the $[0, 1]$ range. Then precipitation values in areas without radar echo are assigned as 0. Finally, we resample the precipitation values on $1024 \times 880$ grids for each radar image, while its actual spatial resolution is about 5 km. After the data preprocessing steps, a 3D matrix with a size of $20,5848 \times 1024 \times 880$ is obtained.





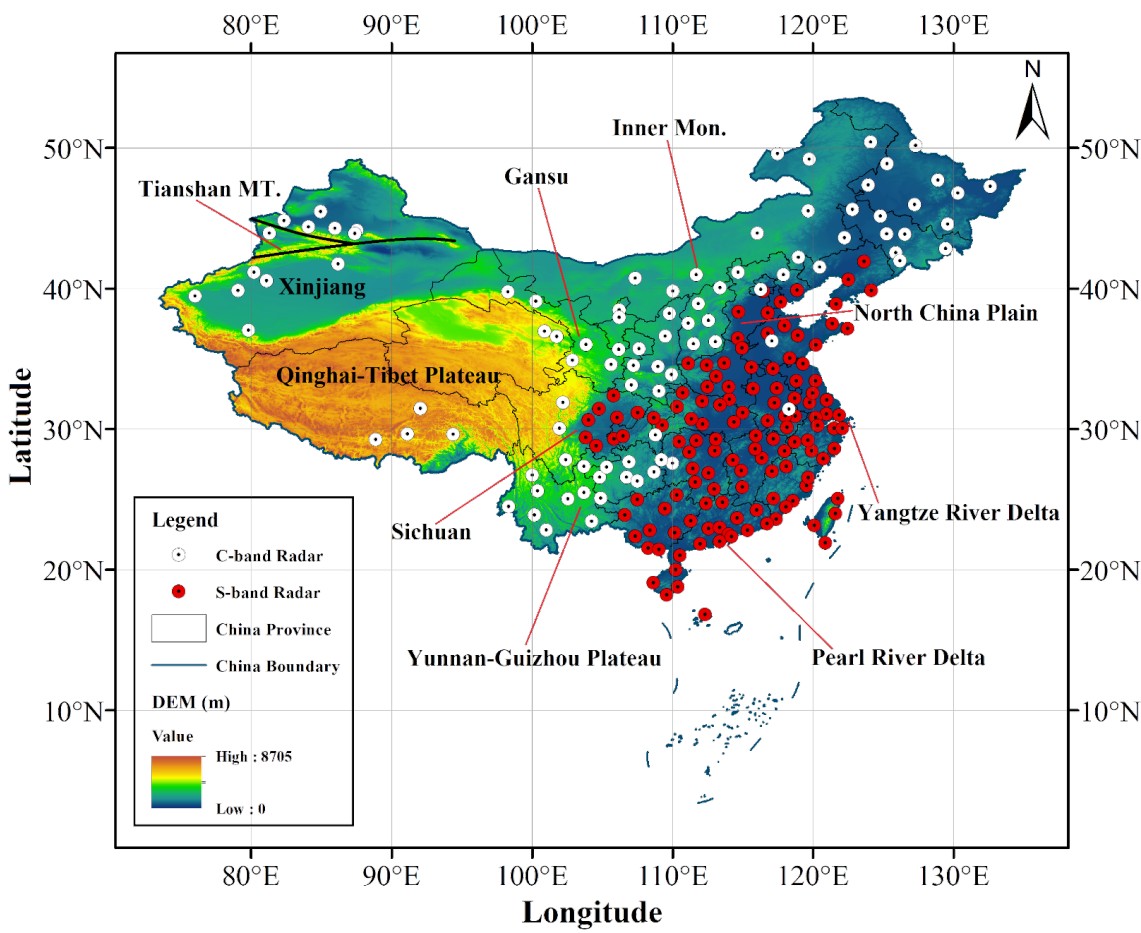

**Figure 1.** The distribution of the CINRAD over China and Topography (unit: m) map. White dots represent C-band radars and red dots denote S-band radars.

## 2.2 Multi-scale Feature Fusion (MFF)

The extrapolative technique of radar echo is an important vehicle for precipitation nowcasting with investigating several key
variables such as the intensity, shape, movement direction, and moving speed of convective cloud. However, there are different
targets (e.g. light rain, moderate rain, and heavy rain) in an echo image or significant differences in the size of the same target
collected at various resolutions. Meanwhile, in a certain region of interest of an echo map, there may be situations of tight
arrangement and disorderly distributions of multiple targets (not least the local strong convection) which inevitably induce
background noises. Therefore, using a single/unique feature (refer to as 'convolution kernel' in a DL architecture) will lead to
low forecast skills due to the relatively small receptive fields.

This study proposes a 3D Multi-scale Feature Fusion (3DMFF, Fig. 2a) module and a 2D Multi-scale Feature Fusion
(2DMFF, Fig. 2b) module. An important part of the 3DMFF is to apply convolution kernels with various sizes to gain dif-





ferent receptive fields. Given the average moving speed of the convective cloud is 36 km/h, the largest convolution kernel with the size of $4 \times 5 \times 5$ could capture the traceability information of the convective cloud under this moving speed. Conversely,

the smallest convolution kernel with the size of $4 \times 2 \times 2$ is geared toward the slow-moving clouds. Besides, a key convolution kernel with the size of $4 \times 1 \times 1$ is also used which is instrumental in dimensionality adjustments and information interactions among channels. The above multi-scale feature are then concatenated so that the module could store more information from the previous echo maps. Similarly, the 2DMFF uses various convolution kernels with sizes ranging from $1 \times 1$ to $4 \times 4$. Furthermore, we introduce the 'Channel-Shuffle' technique (Zhang et al., 2018) to randomly shuffle the concatenated feature

maps along the channel dimensions which enhances the feature interaction ability between channels and further improves the generalization ability of the module. The 3DMFF and 2DMFF module both apply the 'Relu' activation function for nonlinear mapping which thins the network and ease the over-fitting problem to a certain extent.

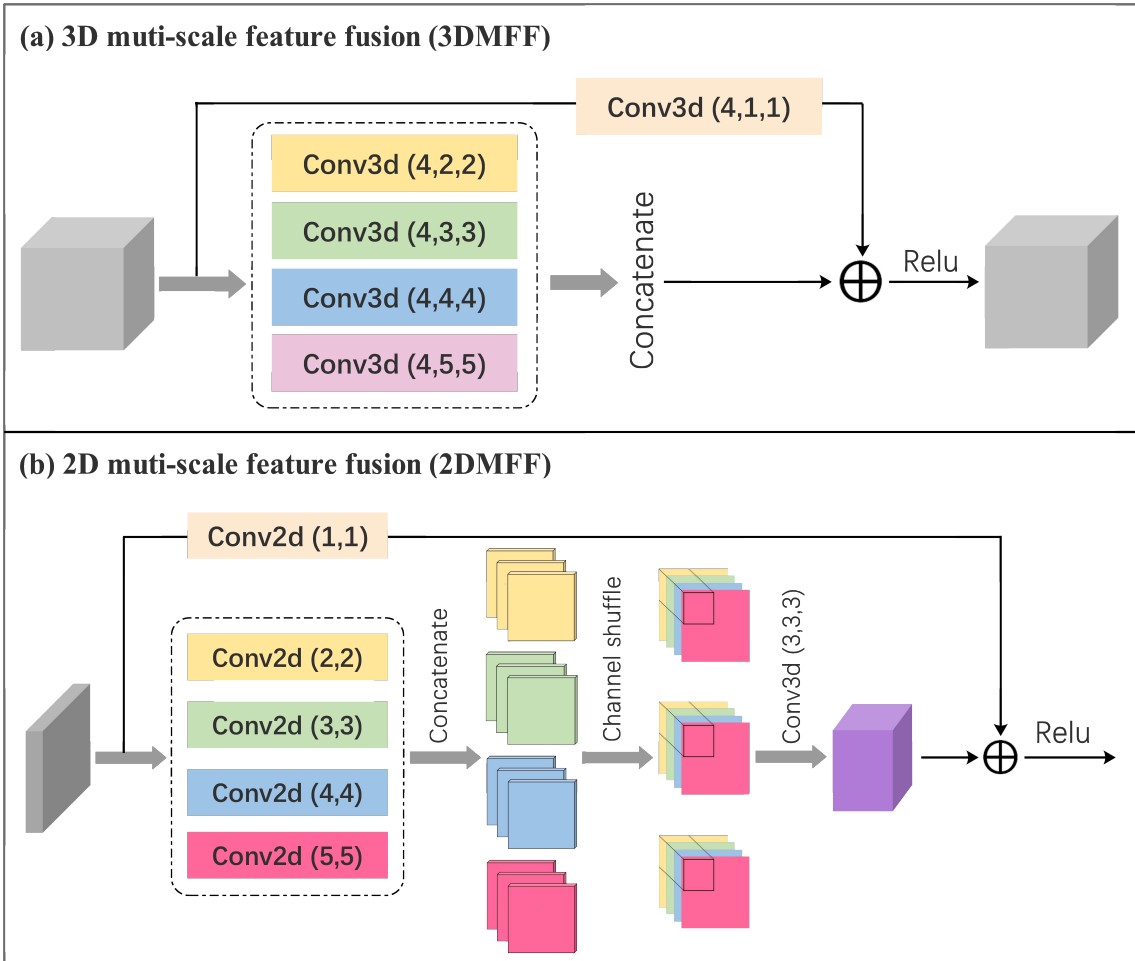

**Figure 2.** (a) The 3D Multi-scale Feature Fusion (3DMFF) and (b) the 2D Multi-scale Feature Fusion (2DMFF).




Consequently, compared to the conventional single feature module, the MFF modules make full use of the information of different receptive fields and enhances the feature interaction ability by increasing the number of network routes. That is

precisely the conventional single feature module is unable to fully extract feature information due to less network routes (refer to as the issue of information loss). In addition, the MFF modules introduce channel sorting and spatial-temporal convolutions to address the issue of information redundancy.

### 2.3    The Framework of the Nowcasting Model based on MFF

We present the complete framework of the precipitation nowcasting model (Fig. 3). Overall, the model uses 60-min radar

echo maps for training with the input size of $(1, 10, 880, 1024)$, and generates the 180-min nowcasting outputs with the size of $(1, 30, 880, 1024)$. The model consists of two steps: the encoding network and the decoding network. Where the encoding network contains a series of down-sampling layers of initial features based on several 3DMFF modules, it plays a role in feature extraction and information compression. As for the decoding network, it applies several 3D transpose convolutions and 2DMFF modules for feature up-sampling and feature restoration. Note that the 3D transpose convolutions also generate a tensor (see $P$

in Fig. 3) which can be deemed as the probability matrix. To fully restore the features of the decoding network to the incipient input's features, two Hadamard product operations are performed: one is the output features of the 2DMFF multiplied by the probability matrix (see $m \odot P$ in Fig. 3) while another is the output features of the 3D transpose convolutions multiply by $1 - P$ (see $(1 - P) \odot f_1$ in Fig. 3). The action of the probability matrix is that it retains the most of intensity information of radar echo as much as possible so that various precipitation systems (e.g. light rain, moderate rain, and heavy rain) can be well

predicted. Because the outputs from 3D transpose convolutions lack edge information (since the use of the padding strategy of 0), so these outputs are also concatenated with the outputs from the 2DMFF modules to reduce information loss. Finnally, we apply a 3D convolution operation to adjust the channel of the product outputs and further generate the precipitation nowcasting results.

By drawing lessons from 'MetNet' (Sønderby et al., 2020), suppose the target weather condition is $y$, and the input condition

is $x$, therefore,

$$p(y|x) = DNN_\theta(x) \tag{1}$$

Where $p(y|x)$ is a conditional probability over the output target $y$ given the input $x$, $DNN_\theta(x)$ is a deep neural network with parameters $\theta$, the model introduces uncertainties due to the calculation of the probability distribution over possible outcomes and does not provide a deterministic output. In most cases, the radar echo reflectivity is a continuous variable, hence we

discretize the variable into a series of intervals and then approximate the probability density function of the variable. Because the discrete probability model can reduce uncertainties, therefore, the combination of discrete probability and radar echo reflectivity will further reduce uncertainties of extrapolative radar echo. Here, we invoke a mechanism of discrete probability as follows:

$$y^{[\tau]} = \Sigma_{i=1}^{c} p_i^{[\tau]}(y^{[\tau]}|x) \cdot x_i \tag{2}$$





Where $y^{[\tau]}$ is the output at time $\tau$, $x$ is the input condition, $c$ is the number of channels, $p_i^{[\tau]}(y^{[\tau]}|x)$ is a conditional probability

at a channel at time $\tau$. Eq. (2) shows the information of multiple channels at a certain time $\tau$. Here, one channel corresponds

to one probability value suggesting that the probability is assigned to each channel to conduct better feature extraction. We

multiply the conditional probability $p_i^{[\tau]}$ by related channel information $x_i$ and then calculate their summation over all channels

so that more realistic radar echo reflectivities are achieved. As can be seen that both $m \odot P$ and $1-P$ (see $(1-P) \odot f_1$ in Fig.

3 use the mechanism of discrete probability.

**Figure 3.** The framework of the nowcasting model based on Multi-scale Feature Fusion (MFF).

Overall, the framework of the nowcasting model with a deep and hierarchical encoding-decoding backbone is instrumental

in extracting the essential features from the inputs, while several plug-and-play modules are suitable to excavate the context in-

formation or meticulous texture features of the inputs and reduce background noises of the inputs, making the model effectively

to investigate the movement vector features of precipitation system (e.g. shape, movement direction, and moving speed) in the

practical nowcasting. Moreover, the model introduces the mechanism of discrete probability to skillfully reduce uncertainties

and forecast errors, making the model postpone the declining rate of strong-intensity echoes to some extent. Therefore, the

model can produce heavy rains with longer lead times.





### 2.4 Comparative Models

To have a comprehensive comparison, three radar echo extrapolation models are also presented here.

#### 2.4.1 Optical Flow (OF)

The radar echo extrapolating problem can be regarded as moving object detection which separates the targets from a continuous sequence of images. Gibson (1979) proposed the concept of OF characterizing an instantaneous velocity of pixel motion of a space object in an imaging plane. Specifically, the OF uses the variation of a pixel of the image sequence in the time domain and the correlation between two adjacent frames, thereby investigating the movement information of objects between consecutive frames. Generally, the transient variation of a pixel on a certain coordinate of the 2D imaging plane is defined as an optical flow vector. The OF method satisfies two basic hypotheses: the grey-scale invariant and the tiny movement of pixels between consecutive frames.

Let $I(x, y, t)$ be the grey-scale value of the pixel at position $(x, y)$ and time $t$, it moves $(dx, dy)$ units of distances using $dt$ units of time. Based on the grey-scale invariant hypothesis, the grey-scale value remains unchanged between two adjacent times, so the following equation holds:

$$I(x, y, t) = I(x + dx, y + dy, t + dt) \qquad (3)$$

Using Taylor expansion, the right term of Eq. (3) becomes:

$$I(x, y, t) = I(x, y, t) + \frac{\partial I}{\partial x}dx + \frac{\partial I}{\partial y}dy + \frac{\partial I}{\partial t}dt + \epsilon \qquad (4)$$

Where $\epsilon$ represents the infinitesimal of the second order which is negligible. Then substitute Eq. (4) into Eq. (3) and divide by $dt$, therefore we have:

$$\frac{\partial I}{\partial x}\frac{\partial x}{\partial t} + \frac{\partial I}{\partial y}\frac{\partial y}{\partial t} + \frac{\partial I}{\partial t} = 0 \qquad (5)$$

Suppose $u = dx/dy$ and $v = dy/dy$ are two velocity vectors of optical flow along the x-axis and the y-axis, respectively. Let $I_x = \frac{\partial I}{\partial x}$, $I_y = \frac{\partial I}{\partial y}$ and $I_t = \frac{\partial I}{\partial t}$ are the partial derivative of the grey-scale of pixels along the x-axis, the y-axis, and the t-axis, respectively. Therefore, Eq. (5) turns into:

$$I_x u + I_y v + I_t = 0 \qquad (6)$$

Where $I_x$, $I_y$, and $I_t$ can be calculated from the original image data, while $(u, v)$ are two unknown vectors. Because Eq. (6) is a constraint equation but has two unknown variables. Therefore, it is necessary to add other constraint conditions to calculate $(u, v)$. Currently, there are two common algorithms used by solving this problem: global optical flow (Horn and Schunck, 1981) and local optical flow (Lucas and Kanade, 1981), detailed mathematical derivations of the two algorithms do not expatiate here.

#### 2.4.2 UNet

The second comparative model is U-Net. The biggest difference between the MFF model and the U-Net model is the latter uses general 2D convolution in place of the 'MFF module'. There are mainly three parts in the U-Net model. The first part is





a backbone network (encoder module) used for down-sampling and feature extraction, and is stacked by several convolution layers and max-pooling layers. Based on the output features from the first part, the second part (decoder module) uses several

up-convolution layers and convolution layers to conduct up-sampling and strengthen feature extraction, so that the features can be fused more effectively. The third part is a prediction module which is used for a specific task such as regression and segmentation. In addition, to ensure the down-sampling feature's size matches the up-sampling feature's size and further reserves more original information, the operation of 'feature copying and cropping' is also needed.

### 2.4.3   Time Series Residual Convolution (TSRC)

The third comparative model is the TSRC model proposed by our previous study (Huang et al., 2023), detailed mathematical derivations of the TSRC are omitted here. The core idea of TSRC is that it compensates the current local radar echo features with previous features during convolution processes on a spatial scale. Moreover, the model implants 'time series convolution' to ease the dependencies on spatial-temporal scales so that more contextual information and less uncertain features are reserved in the hierarchical architecture. Especially, the model exhibits good performance in dealing with the smoothing effect of the

precipitation field and the degenerate effect of echo intensity.

### 2.5   Evaluation Metrics

We utilize five evaluation metrics to examine the forecast skills of the three extrapolative models, including the probability of detection (POD), false alarm rate (FAR), mean absolute error (MAE), radially averaged power spectral (RAPS), and structural similarity index (SSIM).

$$POD = \frac{successful\ forecast}{successful\ forecast + missing\ forecast} \qquad (7)$$

$$FAR = \frac{null\ forecast}{successful\ forecast + null\ forecast} \qquad (8)$$

Where successful forecast, missing forecast, and null forecast typically appear in practical precipitation tasks. The above three values are determined by the comparison between ground true value (GTV), forecast value (FV), and threshold value

(TV). In practical precipitation tasks (Huang et al., 2023), the threshold of 20 dBz represents those reflectivity values greater than 20 dBz (hereafter '~20 dBz'). Similarly, the term '~30 dBz' and '~40 dBz' can be abbreviated. In this study, we adopt three thresholds which are set as 20, 30, and 40 dBz. For example, if GTV ≥ TV and FV ≥ TV, then mark one successful forecast event; if GTV ≥ TV and FV < TV then mark one missing forecast event; if GTV < TV and FV ≥ TV, then mark one null forecast event. Both POD and FAR intuitively describe the performance of growth and dissipation forecasting tasks.

$$MAE = \frac{1}{n}\Sigma_{i=1}^{n}|Y_i^g - Y_i^f| \qquad (9)$$

Where $Y_i^g$ and $Y_i^f$ are the ground truth value and forecast value in the $i-th$ pixel of the related echo image, and n is the total number of pixels. This metric describes the performance of each forecast model at different precipitation intensity levels.





This study regards the grey-scale of radar echo as a signal. The power spectrum describes the magnitude of different signal frequency components of a 2D image, therefore it is treated by the Fourier transform from the spatial domain into the frequency domain (Braga et al., 2014). Different frequency components within power spectra are located at different distances and directions from the base point on the frequency plane. High-frequency components are located more distant from the base point, and different directions from the base point indicate different orientations of the data features. Here, we use RAPS (Sinclair, 2005; Ruzanski, 2011) to investigate the smoothing effect of forecast radar echo maps and discuss the forecast skill on local convection. Detailed mathematical derivations of RAPS are omitted here.

Besides, we calculate the SSIM (Wang et al., 2004) to examine the similarity of precipitation fields between ground true and forecasting radar echo maps.

$$SSIM = \frac{(2\mu_g\mu_f + c_1)(2\sigma_{gf} + c_2)}{(\mu_g^2 + \mu_f^2 + c_1)(\sigma_g^2 + \sigma_f^2 + c_2)} \tag{10}$$

Where $\mu_g$ and $\mu_f$ are the means of ground truth and forecasting radar echo map, $\sigma_g$ and $\sigma_f$ are the related standard deviation, $\sigma_{gf}$ is the covariance, respectively. $c_1$ and $c_2$ are two constants. This metrics reflects the movement of precipitation field between ground truth and forecasting radar echo map.

## 3 Results

### 3.1 Overall Forecast Performances on Testing Data

We use four years' data (2018-2011) for model training and one year's data from 2022 for model testing. Fig. 4 shows the four evaluation metrics POD, FAR, MAE, and SSIM in three reflectivity intervals of $\sim$20, $\sim$30, and $\sim$40 dBz. Overall, POD in the four models consistently plunges with increased forecast lead time for all reflectivity internals, while it is conversely for FAR. The rankings of POD (or FAR) are quite different from the three reflectivity intervals. For example, in $\sim$20 dBz reflectivity, MFF ranks the highest POD during the whole forecast period, it remains stable ranging from 0.6 to 0.8 which is almost twice that of TSRC, OF, and UNet after the 2 h lead time; however, MFF and TSRC hold the nearly equal FAR which are roughly half of that from OF and UNet. In $\sim$30 dBz reflectivity, TSRC ranks the highest POD, followed by MFF; coincidentally, TSRC also ranks the highest FAR before the 1-hour forecast time, while both MFF and TSRC obtain relatively low FAR compared with that of OF and UNet. In $\sim$40 dBz reflectivity, POD in TSRC is ahead of that of the other three models, especially before the 1-hour lead time, and it degrades into that of MFF at the longer lead time; POD in both OF and UNet are lower than 0.2 during the whole forecast period and nearly decline to 0 after 2 h lead time; MFF reports the lowest FAR during the whole forecast period. While the value of FAR climbs from about 0.1 to 0.9; TSRC has a relatively stable FAR, while the value of FAR is higher than 0.5 during the whole forecast period; FAR in OF and UNet rapidly increase from about 0.1 to 0.8 at the first 90 min; FAR in all models are greater than 0.8.

Although MFF produces relatively low POD in high reflectivity ($\sim$30 and $\sim$40 dBz) intervals compared with TSRC, however, it obtains relatively low FAR at the same time. It is evident from the definition of POD/FAR that both more 'successful forecasts' and more 'null forecasts' occur in TSRC, and conversely, fewer 'successful forecast' events and fewer 'null forecast'





events occur in MFF compared with TSRC for high reflectivity intervals. Taking POD=0.6 as a dividing point, it is obvious that MFF yields 'successful forecast' events for the whole forecast period in ∼20 dBz reflectivity; while TSRC, OF, and UNet gain 'successful forecast' events only before 60 min, 24 min, and 36 min, respectively. Similarly, taking FAR=0.5 as a dividing point in ∼40 dBz reflectivity, it can be found that MFF, OF, and UNet report FAR<0.5 only before 2 h, 30 min, and 30 min, respectively, suggesting that at least the three models can avoid half of the 'null forecast' events before 2 h, 30 min, and 30

min, respectively. However, TSRC is unavoidable to produce 'null forecast' events for the whole forecast period in ∼40 dBz reflectivity.

In addition, we examine the MAE and SSIM between nowcasting and ground truth. Overall, the MAE gradually rises with the forecast time for all models. Specifically, in terms of MAE, MFF has the smallest MAE (about 15 dBz) which is ahead of both TSRC and UNet by about 2 dBz reflectivity after 90-min, while OF has the largest MAE, especially in long forecast time.

These suggest that MFF reproduces the precipitation intensity with relatively less overestimation or underestimation compared with the other three models, while OF shows little capacity to do so, especially in a long forecast time. In terms of SSIM, an important finding is that MFF keeps an upward trend, OF enjoys a steadfast position, while TSRC and UNet show a downward trend, especially after 90 min. These indicate that MFF is suitable to capture the shapes of precipitation fields with high with relatively high similarity and its forecast performance increases with forecast times.

Understandably, MFF has a strong ability to dredge the movement vector features of precipitation systems in multi-scale and alleviate the issue of information loss (reduce uncertainties) in high reflectivity intervals. Therefore, by enforcing the mechanism of discrete probability, the model shows favorable superiority especially for high-intensity precipitation systems even at longer forecast times. By using the strategy of compensated information in time series, the TSRC model might struggle to replenish the information on precipitation intensity but inevitably brings the issue of overestimation for the whole forecast pe-

riod, which leads to producing more 'null forecasts' events. OF produces precipitation fields based on the grey-scale invariant and the tiny movement of the precipitation system, so it is difficult to dig the fast changes of precipitation fields, especially in longer forecast time, and also overestimate or underestimate the high-intensity precipitations. UNet performs feature extractions only on the spatial scale which results in information loss and is unable to excavate the fast changes of precipitation fields on the temporal scale and the high-intensity precipitations, therefore, it has poor ability in nowcasting during the whole

forecast period.



**Figure 4.** The forecast results in terms of four evaluation metrics on testing data on the whole year of 2022.





## 3.2 Results of Case Study

Here we show some forecast results of two real precipitation cases (Fig. 5) to further understand the forecast performance of the four models.

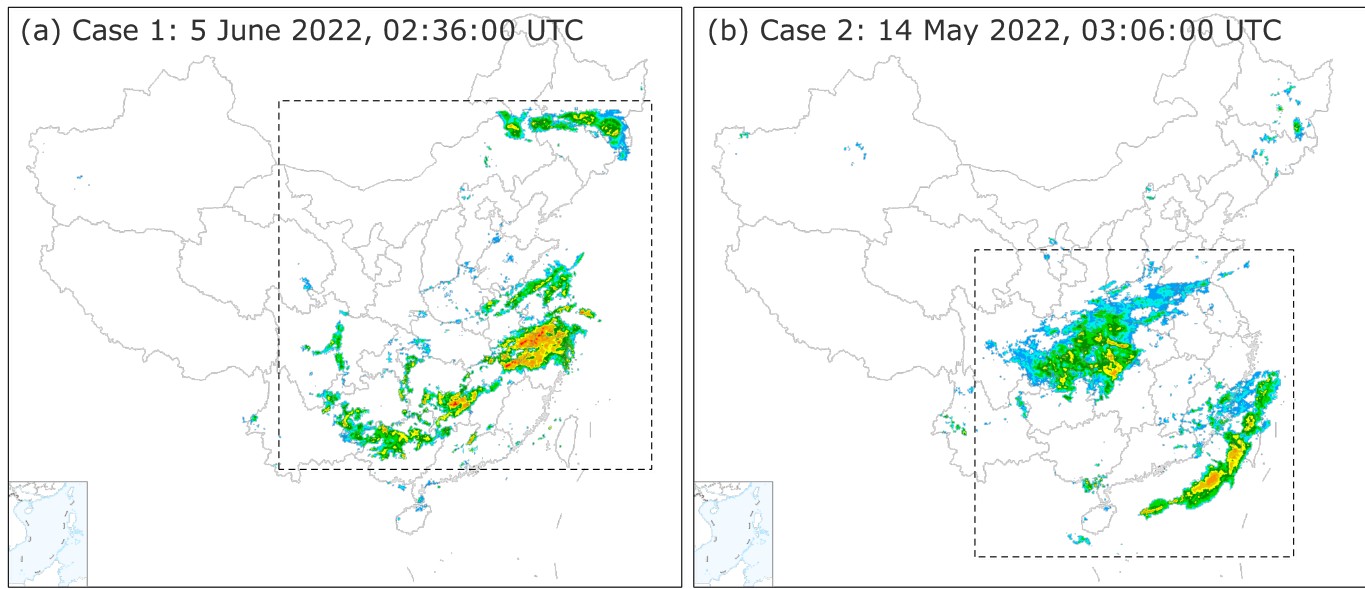

**Figure 5.** The overview of two precipitation cases over China.

### 3.2.1 Case 1

The first case is a large-scale precipitation process over China on 5 June 2025, 02:36:00 UTC (Fig. 5a), it contains the north part over northeastern China affected by a cold vortex and the south part (also known as 'dragon-boat rain') over southern China affected by warm-humid air. Fig. 6 presents the forecast results of this case. In the ground truth (GT), the whole precipitation area keeps a sluggish enlarging trend with the increased lead time, but the precipitation area of the high-intensity (e.g. greater than 35 dBz) echoes narrows gradually. As an important finding, MFF shows the best forecast performances since it can predict
high-intensity (e.g. greater than 45 dBz) echoes even at the longest lead time (T+180 min). Comparatively, TSRC and UNet produce these echoes only at the short-range forecast time and miss them at the longest lead time. In terms of the precipitation field, both MFF and TSRC roughly capture the precipitation area, especially for low-intensity (e.g. less than 30 dBz) echoes at short-range lead times; OF draws an obvious dragged trajectory of the precipitation field in longer lead times, indicating the model simply creates precipitation fields with symbolic replications from the first frame to the last frame (T+180 min)
at a horizontal scale and always misses the local changes of the precipitation system; UNet is definitely difficult to grasp the whole precipitation filed, not to mention the heavy precipitation system and its precipitation filed narrows gradually with the increasing lead time and finally disappears. The above analysis seems to be in accord with previous results that the high POD is reported in MFF and TSRC for low-intensity echoes (Fig. 4a), while the relatively steady POD in OF and UNet for high-





intensity echoes (Fig. 4b and 4c). Overall, MFF outperforms the other three models in predicting the precipitation field and the

heavy precipitation.

**Figure 6.** The precipitation maps of case 1 (5 June 2025, 02:36:00 UTC); the first row: ground truth (GT); the second row: Multi-scale Feature Fusion (MFF); the third row: time series residual convolution (TSRC); the fourth row: optical flow (OF); the fifth row: UNet; the sixth row: the radially averaged power spectral density.





The RAPS is an important metric to intuitively examine the smoothing and blurry precipitation fields, while the lower the power spectral, the smoother the precipitation field is. Conspicuously, OF enjoys a relatively high power spectral which is close to that of ground truth for the whole wavelength range. At first glance OF can predict the meticulous local-convection activity and the evolution of the precipitation system. However, the precipitation field is shifted by the model from the first frame to the
last frame and is to raise 'successful forecast' or 'null forecast' events, indicating the poor forecast performance of the model at longer lead times. The other three DL-based models (MFF, TSRC, and UNet) have relatively low power spectra, suggesting they inevitably introduce a smooth precipitation field to some extent, but they might describe the evolution of the precipitation system more reasonably. Specifically, MFF reports inconspicuous power spectral under 4 km wavelength which is higher than that of TSRC and UNet, while this difference becomes increasingly apparent at longer lead times, suggesting the model has
even more advantages to describe the local-convection activity on a small scale and the model did at least ease the smoothing effect.

### 3.2.2 Case 2

The second case occurred on 15 May 2022, 03:06:00 UTC (Fig. 5b). It was a large-scale precipitation process over central China and offshore of China, and was affected by the upper-level westerly trough, the southwest vortex, and the lower-level
shear. Fig. 7 presents the forecast results of this case. As can be found from the ground truth, the precipitation field gradually expands but the high-intensity area shrinks with lead times. Overall, both MFF and TSRC roughly exhibit the shape of the precipitation areas (on the land and the ocean) and provide the evolutionary trend of the precipitation system. OF still shifts the precipitation field from the first frame to the last frame and certainly misses the evolutionary trend of the precipitation system especially for longer lead times, which further proves the poor ability of the model in long-range precipitation forecasting. As
the precipitation field grows smaller in UNet which is opposite to the ground truth, the model is quite difficult to capture the evolutionary trend of the precipitation system. In terms of echo intensity, the models have different forecasting performances. MFF overestimates these strong-intensity echoes (greater than 30 dBz) but at the same time it enlarges the area of echoes at all lead times; TSRC is unable to produce these strong echo intensities after the 120-min lead time. Although OF can predict these strong-intensity echoes at longer lead times, however, they are almost the same as the first frame which indicates that the
model performs poorly in predicting the evolution of strong-intensity echoes. Unfortunately, UNet shows the worst forecast performance since it underestimates these strong intensities at shorter lead times and can not produce these strong-intensity echoes at longer lead times.

The three DL-based models report relatively low power spectra before the 90 min lead time. OF obtain relatively high power spectra which are almost equal to that of ground truth at all lead times, for the same reason that OF shifts the precipitation filed
by an extrapolative technique. It is noteworthy that the power spectral in MFF is slightly greater than that in TSRC and UNet at longer lead times which suggests that the smoothing effect is further improved by MFF, therefore, this model is more suitable for precipitation forecasting both on the land and the ocean.



**Figure 7.** Same as in Fig. 6, but for the precipitation case 2 (15 May 2022, 03:06:00 UTC).

## 3.3 Discussions

Here we summarize the advantages and disadvantages of the four models in precipitation nowcasting.





### 3.3.1 MFF

The purpose of MFF is to improve the forecast skill of heavy precipitations, especially at longer lead times. Current DL-based models for precipitation nowcasting are facing two challenges: one is poor forecast skills when there are different precipitation systems with various scales; and the other is the low predictive accuracy when various precipitation targets (e.g. light rain, moderate rain, and heavy rain) are densely distributed at a certain area of interest and also introduce noises. From a qualitative perspective, this study proposes MFF which can make full use of the receptive fields to efficiently detect different precipitation systems in multi-scales and predict various precipitation targets. This superiority is unable to be achieved by the traditional single-scale receptive fields. However, while this deep and hierarchical encoding-decoding architecture shows strong ability in feature extraction, it might also account for the issue of information redundancy. Therefore, the model introduces several crafts (e.g. channel shuffle, feature concatenation, spatial-temporal convolution) to enhance the feature interaction ability among multi-scales and further ease information redundancy. The above operations do obtain considerable forecast performances in several evaluation metrics: POD, FAR, MAE, and SSIM. In addition, the model skillfully applies the mechanism of discrete probability which mathematically allocates the probability information into each channel and can reduce uncertainties and forecast errors to the most extent. The results of the case study further prove that only this model can produce heavy precipitations such as those greater than 45 dBz reflectivity radar echoes even at the 3 h lead time. It is noteworthy that two tricky issues (smoothing effect and cumulated error) are still inevitably reported in the model, of course, they are not specific to MFF while most DL-based models are also confronted with the same issues. The principal reasons account for the issues include: the convolution strives to smooth multi-scale feature in receptive fields to minimize fitting errors and the iterative discrepancy between training processes and targets. Encouragingly, by introducing Multi-scale Feature Fusion and the mechanism of discrete probability, at least our models have some improvements which offer much promise for tackling many practical tasks such as precipitation growth and dissipation, fast-moving precipitation system, heavy precipitation, local-convection activity, etc. In any event, MFF is a DL-based and data-driven radar extrapolative model without any consideration of physical constraints and atmospheric dynamics, hopefully, the model can be further improved by combining multi-source data and ingenious DL architectures.

### 3.3.2 TSRC

Essentially, TSRC is a reinforced 'encoding-deconding' architecture, it appends previous features into current feature planes on temporal scales during convolution processes, so more contextual information and less uncertain features could remain in deep networks. The model fully considers the correlation of radar echo features on a temporal scale, therefore, it should theoretically alleviate the problem of information loss and the degenerate effect intensity. However, those compensatory features in the architecture may lack specificity and carry noises, resulting in the model mindlessly increasing the precipitation intensities at the whole forecast lead times. Understandably, the model has relatively high POD though, high FAR and MAE can be found especially for heavy precipitations. Undoubtedly, the model increases the depth of the hierarchical architecture with different learnable parameters, excavates the dependencies of echo features on both temporal scales and spatial scales, and also uses





several crafts (e.g. feature concatenation, residual connection, attention mechanism) to retard the declining rate of intensity and the smoothing effect. The results of testing data show great advantages of the model at those real forecast tasks such as

low-intensity precipitation systems, and slow-change precipitation systems, especially for short lead times. However, due to the fixed/unique receptive field on spatial scales, it lacks consideration of multi-scale feature which leads to great difficulty in many real forecast tasks, such as local-convection activity, growth and dissipation, fast-moving precipitation systems, and rapid changes of rainfall field. This has sparked speculation that the model will be further improved by implementing feature extraction on multi-spatial scales.

### 3.3.3   OF

The core idea of OF is to calculate the change of pixels from the image sequence in the time domain and the dependence in two adjacent frames, and thus investigate the information of moving objects. The forecast results from testing data show that OF is only suitable for the forecast of precipitation systems with slow change at very short lead times. The reason lies in the two basic hypotheses of the model which are the grey-scale invariant and the tiny movement of pixels. For the grey-scale invariant,

it makes the model difficult to deal with that precipitation system with rapid intensity changes. As for the tiny movement of pixels, it can hardly satisfy the forecast of a fast-moving precipitation system. Unlike the DL-based models, OF produces precipitation fields based on Lagrangian persistence and smooth motion which also fail to recognize both the local features of echo and the multi-scales features of echo, resulting in the poor forecasting ability at longer lead times. Therefore, it is easy to understand OF yields forecast results by simply shifting the precipitation fields, while the timeliness of precipitation

forecasting and the accuracy are hard to guarantee. Even with further improved methods such as the semi-Lagrangian method which relies on the advection field, it is still difficult to expound the complex features of the precipitation system.

### 3.3.4   UNet

There are three key steps in the UNet architecture which are encoding, decoding, and skip connection. The encoding part uses several convolution layers for down-sampling and features compression, allowing the contracting path to capture more context

information. Conversely, the decoding part applies several deconvolution layers for up-sampling and feature restoration, allowing the expanding path to locate different features. The skip connection part fuses the pixel-level features and semantic-level features to achieve feature segmentation and reduce information loss. Meanwhile, the bottom of the hierarchical architecture collects low-frequency information in the form of greater receptive fields but fails to capture high-frequency information. Therefore, when confronted with forecast tasks, the model may focus on those global (abstract or essential) features of precip-

itation systems but omit those exquisite changes in precipitation systems at spatial-temporal scales. The radar echoes usually have variability at multi-scales, so it is not insufficient for UNet to capture complex features of the precipitation system. The results from the case study also confirm that the model has poor forecast skills in fast-moving precipitations, high-intensity precipitations, growth and dissipation, and long-term forecasting. In a word, UNet ranks the worst forecast performance among the three DL-based models.



## 4 Conclusions

In this study, we present MFF, a DL-based model for large-scale precipitation nowcasting with a lead time of up to 3 h. The model aims to investigate the movement features of precipitation systems on multi-scales. Moreover, we introduce the mechanism of discrete probability in the model to reduce uncertainties and forecast errors. Three existing radar echo extrapolative models which are TSRC, OF, and UNet are compared with our study. The comprehensive analyses of testing data further prove the impressive forecast skills of MFF under four evaluation metrics POD, FAR, MAE, and SSIM. In addition, from the results of case studies at least, MFF is the only extrapolative model that produces heavy precipitations even at the 3 h lead time, and the smoothing effect of the precipitation field is improved by our models. From an early warning perspective, the model shows a promising application prospect.

It is well known that data always determine the upper limit of a machine learning model, while algorithms only attempt to approximate this limit. Regretfully, the current study only considers the radar echo data as the model inputs. Therefore, it is highly suggested to consider more meteorological variables (e.g., temperature, pressure, humidity, wind, etc.) and ground elevations in future work, while these data come from various sources such as radar observations, satellite sounding, reanalysis, real-time observation, NWP downscaling, etc. We believe these multi-source data would fortify some kind of physical or thermodynamic constraint for a pure data-driven extrapolative model. As one of the thorny problems in most DL-based models, the smoothing effect is still reported as long as the convolution procedure is performed and remains a challenging task. Therefore, future works will focus on the structural adjustments of the network and the combinations with numerical models to further improve the forecast accuracy of heavy precipitations at longer lead times.

*Code and data availability.* The source codes and pretrained models are available on a Zenodo repository https://doi.org/10.5281/zenodo.8105573 (last access: 1 July 2023; Tan, 2023).

*Author contributions.* Conceptualization was performed by JT and SC; QH and JT contributed to the methodology, software and investigation, as well as the preparation of the original draft; QH and SC contributed to the resources and data curation, as well as visualizations and project administration; JT and SC contributed to the review and editing of the manuscript, supervision of the project and funding acquisition. All authors have read and agreed to the published version of the manuscript.

*Competing interests.* The authors declare that they have no conflict of interest.

*Financial support.* This research was funded by GuangDong Basic and Applied Basic Research Foundation (No. 2020A1515110457), the China Postdoctoral Science Foundation (No. 2021M693584), the Guangxi Key R&D Program (No. 2021AB40108, 2021AB40137), the





Innovation Group Project of Southern Marine Science and Engineering Guangdong Laboratory (Zhuhai) (No. 311021001), and the Opening Foundation of Key Laboratory of Environment Change and Resources Use in Beibu Gulf (Ministry of Education) (Nanning Normal University, Grant No. NNNU-KLOP-K2103).

455 *Disclaimer.* Publisher's note: Copernicus Publications remains neutral with regard to jurisdictional claims in published maps and institutional affiliations.

*Acknowledgements.* The authors would like to thank the reviewers for their valuable suggestions that increased the quality of this paper.



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
