# Peer review of "Deep Learning Model based on Multi-scale Feature Fusion for Precipitation Nowcasting"

_Geoscientific Model Development, 2023_

## Author Response (AR1)

**List of Responses**

Dear Editor and Reviewer(s):

Thank you for your letter and the reviewer's comments concerning our manuscript entitled "Deep Learning Model based on Multi-scale Feature Fusion for Precipitation Nowcasting". Those valuable comments are very helpful for revising our paper and improving our future researches. We have polished the manuscript by refining language, editing grammar, and improving word choice. We hope to meet with approval. Once again, thank you and all the reviewers for the kind advice.

**# Broad comments:**

(1) I'm not overly familiar with the method described here, but for other machine learning applications, when a model has to extrapolate beyond the training dataset, it often performs poorly. What have you done here to mitigate that? Does your training dataset encompass the full range of precipitation rates you would ever see? What happens if you have a 1 in 100 year rainfall event? How would the forecast be able to handle that event if it is outside the bounds of the training dataset? Some discussion of these limitations or why they are not limitations is warranted.

**Response:** We are grateful for the suggestion. As suggested by the reviewer, we have added a brief description as follows:

*Furthermore, like other convolution-based DL models, the MFF model also requires highlighting the "inductive bias" to improve its generalization ability. Inductive bias can be thought of as a sort of "local prior". In the case of image analysis, the inductive bias in the MFF model mainly consists of two aspects. The first aspect is "spatial locality", which assumes that adjacent regions in a radar echo image always have relevant precipitation features. For example, the region of strong-intensity echoes is usually accompanied by the region of moderate-intensity echoes. However, this inductive bias may sometimes overestimate the precipitation intensity (see case 2 in Figure 7}). or enlarge the precipitation field, leading to accuracy issues. The other aspect is "translation equivariance", which means that when the precipitation field in the input map is translated, the precipitation field is also translated due to the use of local connection and weight-sharing in the multi-scale convolution process. This feature does allow the MFF model to trace the moving precipitation system. Therefore, as a widely-concerned weather phenomenon, extreme precipitations (e.g. 1/100 year rainfall events) may also be extrapolated and predicted by using inductive bias in the MFF model if both the training dataset and testing input provide precipitation events with very strong radar echoes. Conversely, it is also very challenging for the MFF model to tackle such a forecasting task.*

Please also see the paragraph in Line 393-404.

(2) I agree that your method seems to be doing a better job of capturing the extremes in the precipitation than the other methods presented here, but, particularly for your second case study, the MFF model seems to be overestimating values as compared to the GT. While the smoothing

effect you mention can explain some of this, more discussion about why this overestimation occurs is warranted. From a forecasting perspective, I would think false positives, if they come frequently enough, could be as detrimental as false negatives, as people will start ignoring forecasts if they are always overestimating rainfall. Again, this is a relatively minor issue, but I think a little more nuanced discussion is required.

**Response:** Thank you for your comments. This is a similar question to the previous one, so we presented the related discussion in Line 393-404.

**Specific comments:**

Line 122: I assume you mean the spatial resolution of one radar image (all grid boxes combined) is 5 km? It's unclear from your phrasing. Also, where does the 205848 number come from? Is that the temporal component?

Line 252: Define RAPS.

Line 263: Do you mean 2018 – 2021?

Line 310: Should be 5 June 2022.

**Response:** Thank you very much for your carefulness and patience. We have made corrections to the aforementioned errors. Also, we have polished the manuscript by refining language, editing grammar, and improving word choice. Please also see the manuscript with track changes.